# SSAT-Adapter: Enhancing Vision-Language Model Few-shot Learning with Auxiliary Tasks

## ABSTRACT

Traditional deep learning models often struggle in few-shot learning scenarios, where limited labeled data is available. While the Contrastive Language-Image Pre-training (CLIP) model demonstrates impressive zero-shot capabilities, its performance in few-shot scenarios remains limited. Existing methods primarily aim to leverage the limited labeled dataset, but this offers limited potential for improvement. To overcome the limitations of small datasets in few-shot learning, we introduce a novel framework, SSAT-Adapter, that leverages CLIP's language understanding to generate informative auxiliary tasks and improve CLIP's performance and adaptability in few-shot settings. We utilize CLIP's language understanding to create decision-boundary-focused image latents. These latents form auxiliary tasks, including inter-class instances to bridge CLIP's pre-trained knowledge with the provided examples, and intra-class instances to subtly expand the representation of target classes. A self-paced training regime, progressing from easier to more complex tasks, further promotes robust learning. Experiments show our framework outperforms the state-of-the-art online few-shot learning method by an average of 2.2% on eleven image classification datasets. Further ablation studies on various tasks demonstrate the effectiveness of our approach to enhance CLIP's adaptability in few-shot image classification.

## CCS CONCEPTS

• **Computing methodologies** → **Transfer learning**; **Matching**; *Image representations*.

## KEYWORDS

Vision-Language Models, Few-shot Learning, Auxiliary Learning

## 1 INTRODUCTION

Humans possess a remarkable ability to rapidly learn new concepts from limited examples. After seeing just a few pictures of a stranger, we can easily recognize them within a crowd. This ability involves not only raw computation but also our capacity to combine past knowledge and apply it to novel situations. Few-shot learning aims to replicate this capability in machine learning models, enabling them to adapt to new tasks and domains even with limited labeled data – a scenario where traditional deep learning methods often struggle as they often require vast quantities of labeled data [8, 47].

*ACM MM, 2024, Melbourne, Australia*

© 2024 Copyright held by the owner/author(s). Publication rights licensed to ACM.
ACM ISBN 978-x-xxxx-xxxx-x/YY/MM
https://doi.org/10.1145/nnnnnnn.nnnnnnn

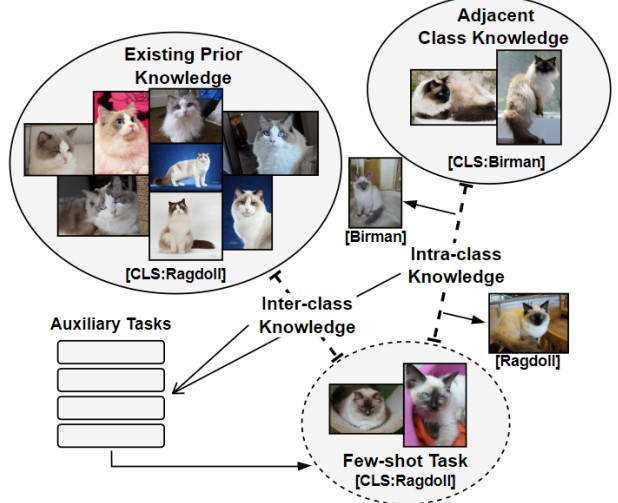

**Figure 1: SSAT-Adapter follows an auxiliary learning approach, learning from additional auxiliary tasks generated from both inter-class and intra-class knowledge.**

The ability of few-shot models to learn from a limited number of examples is paramount in numerous real-world contexts where extensive labeled datasets may be unavailable, costly to acquire, or time-consuming to curate [31, 44]. For instance, few-shot learning techniques have already been applied to several crucial domains such as robotics [40, 53], health [25, 25], and agriculture [3, 54].

The fundamental goal of few-shot learning is to enable models to learn new tasks or concepts from a limited number of labeled examples, with the assumption of the existence of prior knowledge, which is often in the form of pre-trained models [49]. The Contrastive Language-Image Pre-training (CLIP) [43] model represents a significant advancement in machine learning, demonstrating impressive versatility and generalization. Trained on a massive dataset of image-text pairs, CLIP exhibits remarkable zero-shot image classification capabilities. It can categorize images of objects it has never explicitly encountered by leveraging its understanding of natural language descriptions. However, CLIP's performance can still falter when presented with images from novel classes, especially in scenarios where only a few examples are available [62]. This limitation highlights the need for further advancements to enhance CLIP's adaptability in few-shot learning scenarios.

Several strategies seek to enhance CLIP's performance in few-shot learning settings. While fine-tuning the entire CLIP model is possible, it risks overfitting to the limited dataset and demands high computational resources given the size of the CLIP model [51]. Prompt-based methods guide CLIP's classification using natural

language prompts, but designing optimal prompts can be challenging. [18]. More recently, the feature-based approaches [18, 45] have shown promise by introducing lightweight, trainable adapter modules to CLIP. These adapters allow efficient adaptation to new classes while preserving the majority of CLIP's pre-trained weights. However, these methods primarily focus on leveraging the existing dataset, which is inherently limited in few-shot scenarios. While traditional data augmentation techniques like rotations and color shifts might help [7], they do not introduce fundamentally new information the model can learn from. As demonstrated in Figure 1, we hypothesize that augmenting the limited labeled set with existing prior knowledge to generate samples of varying complexities could lead to greater information gains, offering diverse and novel examples that are crucial for model generalization.

To address the challenges of few-shot learning in image classification, we propose a novel framework, SSAT-Adapter, which leverages CLIP's understanding of language and auxiliary learning in a self-paced training regime to obtain additional information and achieve greater few-shot performance improvements. Our approach begins by utilizing the pre-trained CLIP model to generate informative image latents, including anchor instances that represent CLIP's understanding of the target classes.

These anchors, along with other generated inter-class and intra-class latents, create auxiliary tasks designed to boost the performance of the primary few-shot learning task. Inter-class instances bridge the gap between CLIP's pre-trained knowledge and the few available examples. Intra-class instances, created by subtly modifying the latents of labeled examples, expand the representation of target classes. This approach to latent augmentation offers a more compact and computationally efficient representation of visual information. Additionally, while image augmentation can introduce some diversity through random transformations (e.g.,rotations, color shifts), it is often limited in its ability to specifically target the core challenge of few-shot learning – understanding novel classes with minimal data. Latent augmentation allows for more precise manipulation within the learned feature space. SSAT-Adapter transforms latents in specific directions, creating new instances that contain characteristics of the target class. It can also generate points along smooth continuums within the latent space, effectively combining aspects of different classes. This capability to produce a range of diverse and informative instances allows the model to distinguish between similar classes and improve its generalizability in few-shot scenarios. To guide the auxiliary learning model effectively, the training process unfolds in a self-paced manner, starting with easier auxiliary tasks and gradually increasing difficulty as representations move closer to decision boundaries, allowing the model to build a strong foundation by establishing clear initial decision boundaries before refining their placement in relation to more challenging examples, ultimately leading to improved accuracy and adaptability in few-shot learning scenarios.

This paper makes several key contributions to the field of few-shot learning based on pre-trained vision-language models. Firstly, we demonstrate the value of leveraging CLIP's language understanding to inform the generation of auxiliary tasks consisting of informative image latents. Secondly, we introduce the concept of inter-class and intra-class instance generation to subtly expand the

representation of target classes. Finally, we demonstrate the importance of self-paced training for generated instances, moving from easier to more complex auxiliary tasks, and promoting more robust learning even with minimal data. By combining CLIP's pre-trained knowledge, targeted data generation, and self-paced training, our framework enables models to achieve superior accuracy and adaptability in challenging few-shot scenarios.

In the following sections, we first discuss related literature in Section 2. We then establish the required background and problem formulation in Section 3. Section 4 provides a detailed explanation of our proposed framework. Section 5 presents comprehensive experimental results, demonstrating our framework's performance against existing CLIP-based few-shot learning methods across diverse datasets. To further validate our approach, we also conduct extensive ablation studies that highlight the impact of individual components within our framework. Finally, Section 6 summarizes our findings and outlines promising future research directions.

## 2 RELATED WORK

Advances in large-scale pre-trained models within natural language processing (NLP) have paved the way for extensive work on vision-language models (VLMs), such as VisualBERT [29], OSCAR [30], Uniter [9], often utilize transformer-based architectures like BERT [14] for language encoding. More recently, Contrastive Language-Image Pre-training (CLIP) [43] and similar models [27, 28] have demonstrated the power of contrastive learning for visual language tasks. CLIP trains two neural network-based encoders using a contrastive loss to match corresponding image and text pairs, resulting in remarkable zero-shot image recognition capability.

Adapting VLMs for new tasks, especially in few-shot scenarios, is essential for their real-world application. While fine-tuning the entire VLM can be effective, it risks overfitting to limited data and incurs high computational costs [51]. Prompt-based methods offer a compelling alternative by reframing tasks as "fill-in-the-blank" problems, leveraging a pre-trained language model's knowledge. Techniques like Context-Optimization (CoOp) [63] and Conditional Context Optimization (CoCoOp) [62] replace hand-crafted templates with optimizable continuous prompts. Prompt design, whether manual or automated [11, 26, 48, 57], is crucial for the success of these methods. Feature adapter methods [18, 45, 58] have also proven effective in few-shot image classification. They integrate few-shot knowledge with CLIP's pre-trained representations by introducing lightweight, trainable modules that operate on the VLM's output while keeping CLIP's parameters frozen. Although powerful, these approaches primarily focus on leveraging the existing, inherently limited few-shot dataset. Our framework takes a novel approach by expanding beyond the constraints of the existing dataset through targeted boundary data generation.

Data generation techniques [1, 6, 16, 22] can augment the limited examples in few-shot scenarios. While traditional image transformations offer some benefits, they do not fundamentally expand the information the model learns from [5, 23]. Our work focuses on latent-space image generation specifically informed by CLIP's language understanding. This targeted approach introduces novel and decision-boundary-focused training examples, enhancing the model's ability to learn and generalize from limited data.

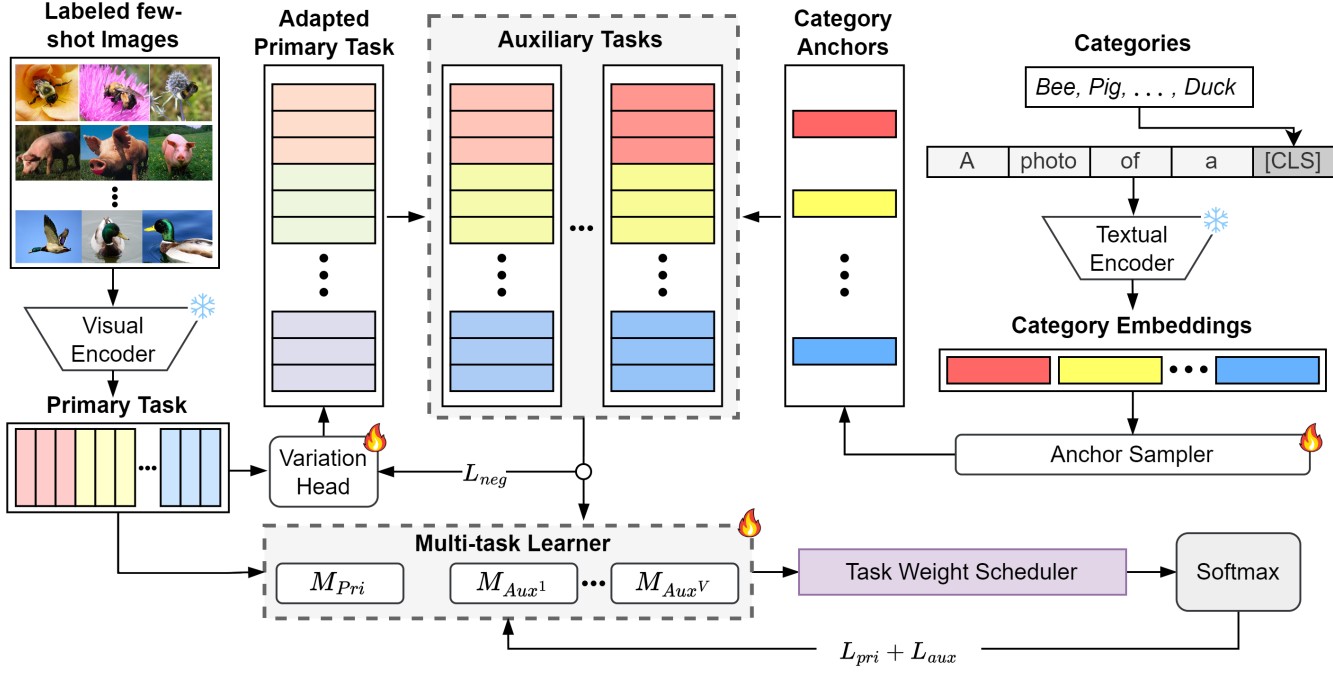

**Figure 2: Overview of the proposed SSAT-Adapter framework, which leverages the pre-trained knowledge of the frozen CLIP visual and textual encoders to generate informative class anchors and diverse auxiliary tasks to assist in learning the primary few-shot task. Trainable and frozen models are annotated with a "fire" symbol and an "ice" symbol, respectively.**

Auxiliary learning leverages the idea of multi-task learning [60], where a model is trained on both a primary task and one or more auxiliary tasks. The auxiliary tasks are designed to complement the primary goal, either by providing supplementary information or introducing regularization to improve generalization [33]. Auxiliary learning has been successful in various machine learning domains [21, 50, 59], including few-shot learning [2, 10, 37] and various applications [34, 38, 55]. To the best of our knowledge, this research is the first to explore the potential of auxiliary learning specifically for adapting vision-language models in few-shot settings.

Self-paced learning, which gradually increases task difficulty during training, provides clear initial decision boundaries for the model and allows it to progressively adapt to more challenging examples. Self-paced learning has demonstrated success in various domains, including self-supervised contrastive learning [32, 42, 42], meta-learning [56], object detection [19], and domain generalization [61]. Nonetheless, the effectiveness of self-paced learning in few-shot learning remains largely unexplored.

## 3 PRELIMINARIES AND PROBLEM FORMULATION

We first review the basic framework for image classification using pre-trained VLMs: Given an image $I \in \mathbb{R}^{H \times W \times 3}$ (where $H$ and $W$ denote height and width, respectively), a neural network backbone transforms the image into a feature vector $f(I) \in \mathbb{R}^D$ within a $D$-dimensional feature space. To perform classification, the image feature vector is then multiplied by a classifier weight matrix $W \in \mathbb{R}^{D \times K}$ (where $K$ is the number of classes) to obtain a

$K$-dimensional logit. Finally, a softmax function converts the logit into a probability distribution $p \in \mathbb{R}^K$ predicting the likelihood of the image belonging to each of the $K$ classes. In contrast to supervised learning with large amounts of data, we are interested in image classification using few-shot examples. Directly fine-tuning the neural backbone and classifier from scratch in the few-shot setting risks overfitting and often suffers from severe performance drops on the test split. A common approach is to first pre-train the backbone on a large-scale dataset and then knowledge transfer to downstream tasks, either by zero-shot prediction or further fine-tuning on the few-shot examples.

CLIP adheres to the zero-shot transfer paradigm. It pre-trains both visual and textual encoders using contrastive learning on large-scale noisy image-text pairs, aligning image and text representations in a shared embedding space. For image classification tasks, CLIP constructs prompts from category names $C_i$ (e.g., "A photo of a [CLS]"). The text encoder transforms the textual descriptions into classifier weights, allowing direct prediction without fine-tuning. Specifically, given an image classification downstream dataset that contains $K$ categories with their natural language name $C_1, \ldots, C_K$, CLIP inserts each category name $C_i$ into a pre-defined hard prompt template $H$. Then, the language feature extractor encodes the resulting prompt as a classifier weight $\mathbb{W}_i$, which can then be used to perform classification.

CLIP-adapter [18] enhances CLIP's few-shot classification capabilities by introducing a small, learnable adapter module on top of the frozen CLIP backbone. Let's denote the CLIP image encoder as $f_{CLIP}(\cdot)$. The CLIP-adapter module $A(\cdot)$ is a lightweight linear

layer. The adapted feature representation is then calculated using a residual connection: $f_{adapted}(I) = f_{CLIP}(I) + A(f_{CLIP}(I))$. During fine-tuning, only the parameters of the adapter module $A(\cdot)$ are updated, while the original CLIP backbone remains fixed. For both prompt-based methods and fine-tuning, adapting to new classes involves adjusting either classifier weights $\mathbb{W}_i$ or image latents, which are essential for calculating prediction probabilities.

# 4  SSAT-ADAPTER

While prompt-based methods offer advantages, we adopt a fine-tuning approach within our SSAT-Adapter framework to prioritize adaptability and finer-grained control of the pre-trained CLIP image space. However, due to the enormous parameter space of CLIP and the scarcity of training examples in a few-shot setting, fine-tuning the entire backbone is computationally expensive and prone to overfitting. Therefore, similar to that of CLIP-adapter, we only append a small number of additional learnable bottleneck linear layers to CLIP's image encoder while keeping the original CLIP backbone frozen during few-shot fine-tuning. To further enhance the learning process and gain additional information from limited labeled data, we aim to generate both intra and inter-class instances to gain additional information and improve model robustness. An overview of the SSAT-Adapter is shown in Figure 2.

*4.0.1  Class Anchor Generation.* Specifically, given the input image $I$, the corresponding label $C_I$ and a set of categories' natural language names $C_{i=1}^K$, the image feature $f(I)$ and classifier weight $\mathbb{W}$ from the original CLIP backbone are computed. With the classifier weights, class anchors denoted by $A_I$ are first generated. Class anchors are dynamically generated image representations that reflect the key features of an input image's ground true class. The class anchor is generated by rejection sampling of a feature manifold $f \in R^D$ that minimizes the acceptance loss $L_{anchor}(A_I)$, and is accepted if $L_{anchor}(A_I) > \delta$, where $\delta$ is the anchor acceptance threshold. Rejection sampling helps to ensure that the generated class anchor is both highly representative of the true class with respect to $\delta$ and distinct from other classes. The acceptance loss $L_{anchor}$ is a measure of the suitability of a generated sample as a class anchor taking into account both the class probability as well as the diversity of the class probability distribution, defined by:

$$L_{anchor}(A_I) = |\hat{p}_{C_I} \cdot \mathcal{D}(A_I) - t| \qquad (1)$$

where $\hat{p}_{C_I}$ is the predicted probability of the image $I$ belonging to its ground truth class $C_I$ and $t$ is a acceptance boundary width, and $\mathcal{D}(A_I)$ is the diversity score of the anchor $A_I$, calculated as

$$\mathcal{D}(A_I) = \frac{\sigma(S(A_I, C))}{\sqrt{(K-1)/K}}, \qquad (2)$$

$(S(A_I, C))$ represents a vector of similarity scores between the anchor $A_I$ and the set of natural language class names $C_{i=1}^K$.

*4.0.2  Auxiliary Task Generation.* To augment the limited labeled data and guide the learning process, SSAT-Adapter generates a series of image representations through linear extrapolation. Linear extrapolation is carried out at $V$ equal intervals between the class anchor $A_I$ and the original image feature $f(I)$. This results in a set of extrapolated image features $f_A = \{f^1(I), \ldots, f^V(I)\}$. We denote each of these extrapolated features as an auxiliary task,

providing supplementary information to help the primary task of classifying the original image. Specifically, for each extrapolated image $f_A$, a initialized auxiliary linear models with residual connections $\{M_{aux}^1, \ldots, M_{aux}^V\}$. Additionally, a primary model, $M_{pri}$, is constructed as the learner for the main classification task of the original image feature $f(I)$. The extrapolated image features can be seen as variations of the original image, positioned along a spectrum between the 'ideal' class representation (the class anchor) and the unaltered input image. These variations offer diverse perspectives to the learning process.

*4.0.3  Auxiliary Learning.* SSAT-Adapter leverages both auxiliary and primary models to enhance learning from the limited data in a few-shot setting. Auxiliary models $(M_{aux}^1, \ldots, M_{aux}^V)$ process the extrapolated image features generated during linear extrapolation. Each auxiliary model focuses on a slightly different variation of the original image, positioned between the class anchor and the original image feature. By learning from these variations, the auxiliary models help capture diverse aspects of the target class. The primary model's objective is to classify the original image. It processes the image feature $f(I)$ directly, leveraging both the pre-trained knowledge of CLIP and the fine-tuning guided by the extrapolated instances and their associated auxiliary models. Given the set of auxiliary models and the primary model, the learned primary latent and auxiliary latents are given by:

$$f_{pri}(I) = \alpha M_{pri}(f(I)) + (1 - \alpha)f(I), \qquad (3)$$
$$f_{aux}^v(I) = \alpha M_{aux}^v(f^v(I)) + (1 - \alpha)f^v(I). \qquad (4)$$

where $\alpha$ is the residual ratio, controlling the balance between the model's output and the original or extrapolated feature. Based on these outputs, the category probability vector for each task (both primary and auxiliary tasks) is calculated, denoted by $P^v = \{p_i^v\}_{i=1}^K$ for task $v$. The final prediction within each task $v$ is made by selecting the class with the highest probability: $\hat{i}_v = \arg\max_i p_i^v$.

*4.0.4  Self-paced Task Weighting.* To effectively guide the learning process in a few-shot setting, SSAT-Adapter employs a self-paced task weighting scheduling strategy that combines a weighted cross-entropy loss with dynamic weight adjustments for the auxiliary models. During training, the parameters of both the primary and auxiliary models are optimized. The loss function is a weighted sum of the cross-entropy losses across all tasks, including the primary task $L_{pri}$ and tasks corresponding to the extrapolated features $L_{aux}$:

$$L(I) = -\sum_{v=1}^{V}\sum_{k=1}^{K} w_{kv} y_I^k \log(p_I^k) \qquad (5)$$

The weights assigned to each auxiliary model evolve throughout the training process. Initially, higher weights are assigned to auxiliary models associated with extrapolated instances closer to the class anchor $A_I$. Recall that the class anchor represents the input's true class with respect to $\delta$. The weights gradually shift as training progresses to dynamically adjust the weights for each of the auxiliary models with respect to training epochs. Specifically the weight of a model $w_{kv}$ for class $k$ in task $v$ at epoch $e$ is given by:

$$w_{kv}(e) = w_{kv}^{initial} + (w_{kv}^{final} - w_{kv}^{initial}) \cdot \frac{e}{E_{max}}, \qquad (6)$$

where $w_{kv}^{\text{initial}}$ and $w_{kv}^{\text{final}}$ is the initial weight and final weight respectively, and $E_{max}$ is the predefined maximum number of epochs where $w_{kv}(e) = w_{kv}^{\text{final}}$. Using the weight scheduler, higher weights are given to auxiliary models with extrapolated instances that are closer to the generated class anchor $A_I$, and lower weights are given to auxiliary models with extrapolated instances that are closer to the input image $I$. Focusing on these tasks early in training helps the model build a strong knowledge foundation of distinct initial decision boundaries, allowing for the introduction of more difficult tasks at a later training stage. As the training epoch progresses, the model weight shifts to focus on auxiliary models that are closer to the input image $I$ with less weight given to auxiliary models with instances closer to the generated class anchor $A_I$. This shift in focus helps to refine the model's ability to distinguish subtle differences between classes and make accurate predictions on the few-shot image classification task.

*4.0.5 SSAT-Adapter with Variation.* To further enhance learning in few-shot scenarios, we introduce a variation to SSAT-Adapter that focuses on difficult instances lying closer to decision boundaries between classes. By concentrating on these challenging cases, we aim to maximize information gain and improve the model's ability to discriminate between subtle differences, reducing overfitting. Specifically, we utilize two key mechanisms to generate more challenging instances focused near decision boundaries. First, when generating class anchors $A_I$, the acceptance threshold $\delta$ is decreased. This modification leads to class anchors that are less 'ideal' representatives of their class. They exhibit lower similarity to the natural language representation of the image's true class and have a more uniform class probability distribution, indicating proximity to decision boundaries. Secondly, the variational head model $M_{var}()$, a linear layer with residual connection, adapts the primary latent representation. The goal is to transform the original image feature into a modified version $I_{var} = M_{var}(f(I))$, employing a modified contrastive triplet loss function. This loss function promotes targeted adaptation by moving the primary instances away from their original CLIP anchors and encourages focused learning by considering the proximity to the nearest instance from a different class. We first denote the nearest latent representation from a different class as $f_{neg}(I)$, representing a challenging "negative" example. The triplet loss for our variation head model can then be formulated as:

$$L_{neg}(I) = d(f_{neg}(I), I_{var}) - d(f(I), I_{var}), \tag{7}$$

where $d(\cdot, \cdot)$ is the euclidean distance measure between representations. To determine $f_{neg}(I)$, the euclidean distance between the adapted latent $I_{var}$ and the latent representations of images belonging to classes other than the ground truth class of image $I$ is measured, the instance with the smallest distance is denoted as $f_{neg}(I)$. This adapted representation lies closer to the decision boundaries, making the classification task more difficult. By adapting both the class anchor $A_I$ and the primary latent $f(I)$, the extrapolated instances generated between them also lie in this challenging region near decision boundaries. These extrapolated instances provide valuable auxiliary information, bridging the gap and enhancing the learning process.

# 5 EXPERIMENTS

## 5.1 Dataset

To evaluate SSAT-Adapter's performance, we adopt a methodology similar to CLIP [43] and CLIP-adapter [18]. We utilize 11 diverse image classification datasets: Caltech101 [17], DTD [12], EuroSAT [20], FGVCAircraft [36], Food101 [4], OxfordFlowers [39], ImageNet [13], OxfordPets [41], StanfordCars [24], SUN397 [52], and UCF101 [46]. Our SSAT-Adapter is trained under few-shot setups (1, 2, 4, 8, 16 shots) and evaluated on full test splits. For datasets without predefined train-test splits, we create a 50/20/30 train/validation/test split. K-shot instances are sampled from the train split, and testing occurs on the full test split.

## 5.2 Training Settings

We employ ViT-B/32 [15] as the visual encoder and BERT [14] as the textual encoder for the CLIP backbone in most experiments. We set the hidden embedding dimensionality of both the visual and text bottleneck layers to 256. SSAT-Adapter optimization occurs on the training set with a batch size of 32, using the AdamW optimizer [35] and a learning rate of 0.0001. Our framework relies on three key hyperparameters: the anchor acceptance threshold, the max weight epoch, and the residual ratio. We conduct hyperparameter searches across different value selections for each dataset, reporting the best performance within the search spaces. Following the approach of CLIP-adapter, we utilize the same prompt template. This template consists of hard prompts like "a photo of a CLS" for generic image datasets, and more specific prompts for fine-grained classification (e.g., "a photo of a CLS, a type of flower" for OxfordFlowers). All the source code are available at https://anonymous.4open.science/r/ACMMM24-F58C/.

## 5.3 Baselines

We benchmark SSAT-Adapter against three baseline models: Zero-shot CLIP [43], Meta-Adapter [45], and CLIP-Adapter [18]. To ensure a fair comparison, we use the same prompt template across Zero-shot CLIP, CLIP-Adapter, and SSAT-Adapter. Meta-Adapter takes the textual category embeddings from pre-trained VLMs and refines them using the information from the few-shot image samples. This is achieved through a gated multi-head attention mechanism. The mechanism allows the model to selectively focus on relevant parts of the image features while processing the category embeddings. As discussed previously, CLIP-Adapter introduces trainable modules that operate on the VLM's output while keeping CLIP's parameters frozen. Given that there are many variants of CLIP-Adapter, we adopt the best performing CLIP-Adapter variant for our experiments, which features additional residual linear layers exclusively on the visual encoder.

## 5.4 Performance Comparison & Analysis

Figure 3 demonstrates SSAT-Adapter's consistent performance advantage over Zero-shot CLIP, Meta-Adapter, and CLIP-adapter across all datasets and few-shot settings. As expected, the most accuracy gains occur in scenarios with limited labeled data (1-shot and 2-shot), showcasing SSAT-Adapter's effectiveness in leveraging auxiliary tasks and self-paced learning, with less accuracy gain

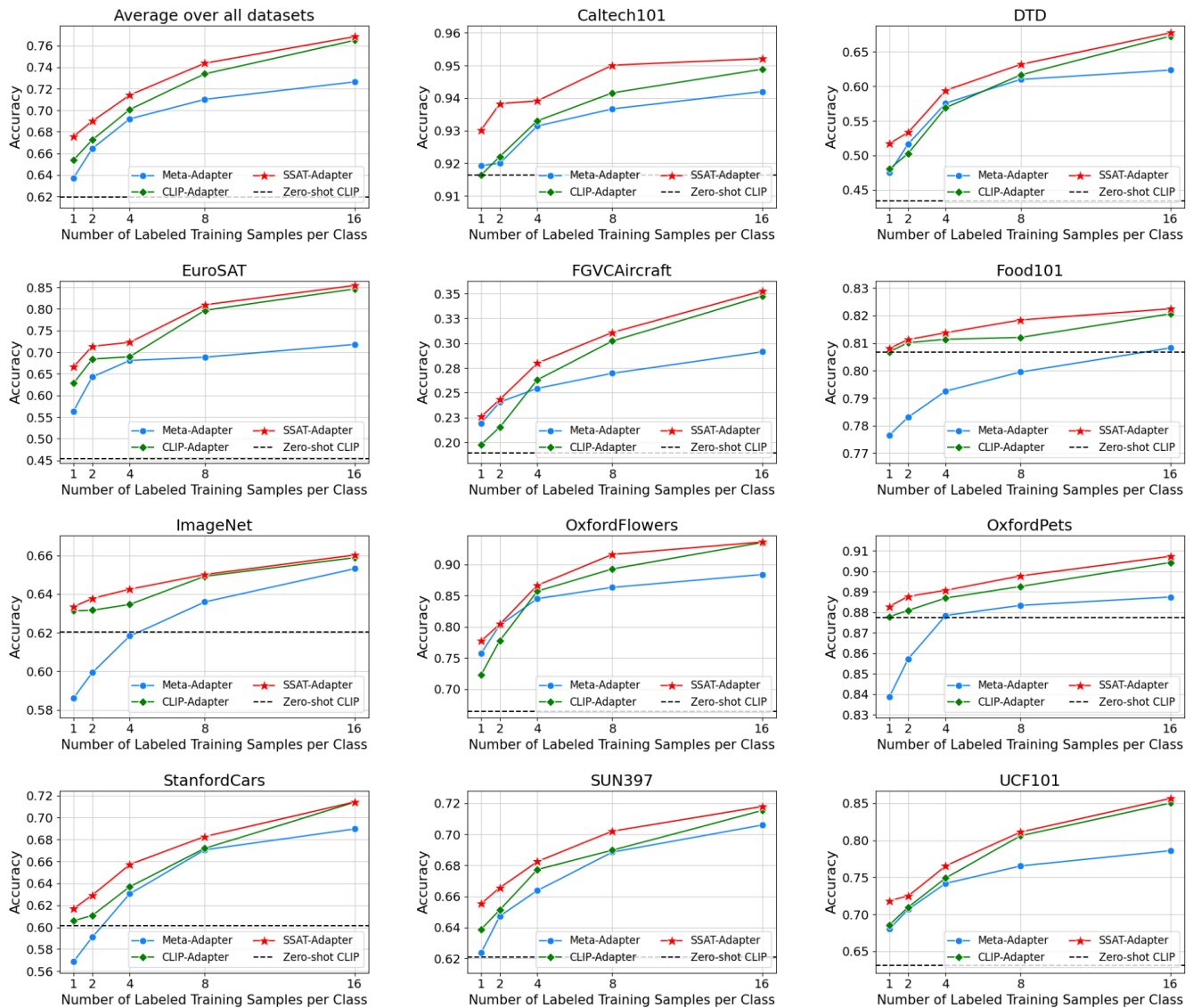

Figure 3: SSAT-Adapter demonstrates superior few-shot learning accuracy across 11 datasets, outperforming previous baselines at various training shot levels. The average accuracy across 11 datasets is shown on the top left.

when the amount of labeled training samples increases. Compared to Zero-shot CLIP [43], SSAT-Adapter achieves significant accuracy improvements over all 11 datasets. The highest accuracy gains are observed particularly on datasets with intrinsically low zero-shot performance like EuroSAT. The accuracy gain is smaller for more complex and generic datasets that contain instances from disjointed classes, such as ImageNet and dataset that already have high zero-shot accuracy, such as the OxfordPets and Food101 dataset. Compared to Meta-Adapter [45], SSAT-Adapter also shows comprehensive performance advantages. For many datasets, under 1-shot and 2-shot training setups, Meta-Adapter barely reaches the accuracy of Zero-shot CLIP, but SSAT-Adapter can always surpass Zero-shot CLIP and exceed Meta-Adapter. It is also noticeable that

SSAT-Adapter also consistently achieves a significant accuracy gain over Meta-Adapter even when the number of labels training samples is high (16-shot), achieving a 4.2% average accuracy improvement across all 11 datasets. The most accuracy gain is also observed on the EroSAT dataset, with an accuracy gain of 10.3% and 13.7% at 1-shot setting and 16-shot setting, respectively. Compared to CLIP-adapter [18], which has already gained huge improvements over Zero-shot CLIP and Meta-Adapter, SSAT-Adapter still outperforms CLIP-Adapter on all datasets under 1-shot and 2-shot training setups. Both methods achieve a similar performance under 16-shot training setups. On average, SSAT-Adapter achieves 2.2% and 0.3% accuracy increase at 1-shot setting and 16-shot setting

respectively, with the highest accuracy gain of 5.4% observed for the Oxfordflowers dataset under 1-shot setting.

## 5.5 Ablation Studies

In this section, we report a series of detailed ablation studies to validate the effectiveness of each design in SSAT-Adapt. We provide 4 experiments to provide additional insight on: 1) The effectiveness of the variation head model. 2) The effect of different self-paced training strategies. 3) The choice of loss function for the adaptation head model. 4) The performance gain from auxiliary tasks.

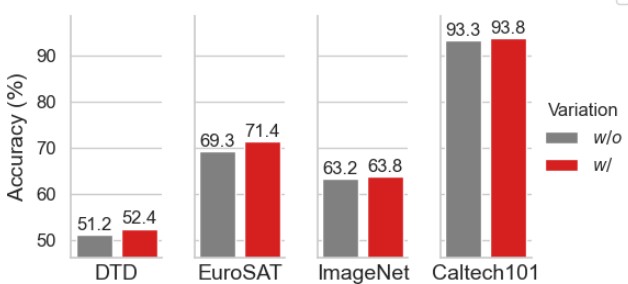

**Figure 4: Comparison of SSAT-Adapter with ($w/$) or without ($w/o$) variation head model.**

*5.5.1 Effectiveness of Variation Head.* The goal of the variational head model $M_{var}()$ is to adapt the primary latent representation to be closer to the decision boundaries. This adaptation allows for the construction of more difficult auxiliary tasks and increases model robustness. Figure 4 demonstrates the performance advantage of SSAT-Adapter with the variation head across four datasets. The results show that SSAT-Adapter with the variation head consistently outperforms the model without it across all datasets.

Additionally, we notice a higher accuracy improvement for DTD and EuroSAT in comparison to ImageNet and Caltech101. Figure 5 shows the pairwise class similarity comparison between the datasets. The similarity results suggest DTD and EuroSAT are more specific datasets that exhibit a more uniform underlying data distribution within each class (e.g., DTD contains textures, EuroSAT includes satellite imagery). In such cases, class instances are inherently similar, making boundary cases especially critical for accurate classification. By adapting the representation towards the decision boundaries, the variation head improves the model's ability to handle these boundary instances, leading to an overall boost in accuracy. Conversely, while we still observe increased accuracy for ImageNet and Caltech101 when using the variation head, the magnitude of improvement is less significant. These datasets feature a broader range of classes with inherently more distinction between them. Consequently, boundary cases are less frequent, and the variation head has a comparatively smaller impact.

*5.5.2 Task Weight Scheduling.* To analyze the impact of self-paced training, we experimented with three task weight scheduling strategies for our auxiliary tasks. The first method serves as a baseline, using equal weights for all auxiliary tasks throughout training. The second strategy prioritizes difficult auxiliary tasks initially, with

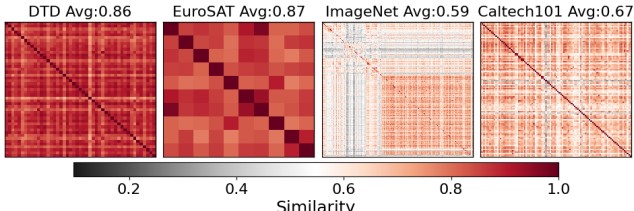

**Figure 5: Pairwise class similarity comparison between specific datasets (EuroSAT, DTD) and generic datasets (ImageNet, Caltech101).**

**Table 1: Comparison of SSAT-Adapter with different task weight scheduling methods under 2-shot training setting.**

| Dataset | Equal | Hard → Easy | Easy → Hard |
|---|---|---|---|
| DTD | 0.504±0.002 | 0.489±0.001 | **0.524±0.001** |
| EuroSAT | 0.693±0.001 | 0.670±0.001 | **0.714±0.001** |
| ImageNet | 0.611±0.002 | 0.603±0.002 | **0.638±0.001** |
| Caltech101 | 0.922±0.001 | 0.920±0.002 | **0.938±0.001** |

weights gradually decreasing in favor of easier tasks. Conversely, the third strategy adopts a "start easy, end hard" approach, initially emphasizing easier tasks and progressively increasing the weights for difficult ones. Table 1 demonstrates the importance of a self-paced strategy with an initial emphasis on easier tasks. This approach allows the model to establish a strong foundation of clear decision boundaries by learning the core concepts and features relevant to the main task allowing for the adaptation of complex concepts. In contrast, the initial focus on difficult tasks hinders the model's ability to grasp fundamental concepts effectively, leading to weaker overall accuracy. These results suggest that gradually increasing the difficulty of auxiliary tasks within a self-paced training regime can significantly benefit model performance in few-shot learning scenarios.

**Table 2: Comparison of SSAT-Adapter with different adaptation strategies under 2-shot training setting.**

| Dataset | Adaptation Strategies | | | |
|---|---|---|---|---|
| | Furthurest | Random | Average | Nearest |
| DTD | 0.523±0.002 | 0.512±0.012 | 0.516±0.002 | **0.524±0.001** |
| EuroSAT | 0.712±0.002 | 0.711±0.008 | 0.711±0.001 | **0.714±0.001** |
| ImageNet | 0.636±0.001 | 0.634±0.004 | 0.633±0.002 | **0.638±0.001** |
| Caltech101 | 0.936±0.001 | 0.934±0.006 | 0.935±0.001 | **0.938±0.002** |

*5.5.3 Adaptation Strategies.* Our goal is to expand target class representations and obtain new knowledge for improved few-shot learning. To achieve this, we investigate several adaptation strategies for the adaptation head model, each focusing on manipulating few-shot instance representations in distinct ways. Figure 6 presents an illustration of the four adaptation strategies. Our first strategy,

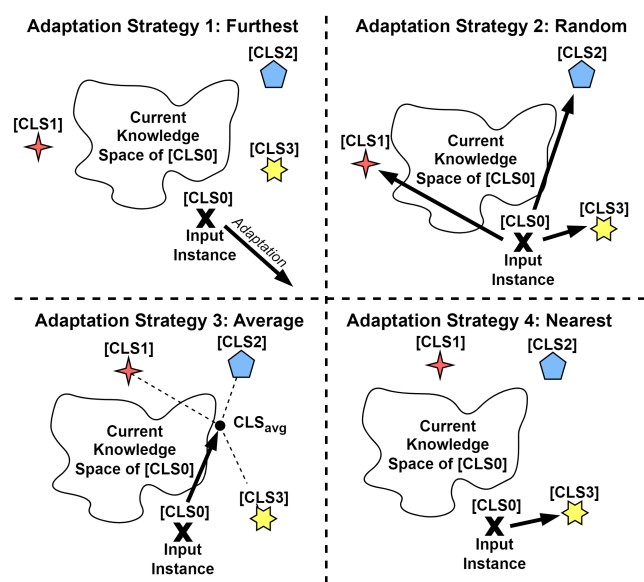

**Figure 6: Demonstration of four different adaptation strategies for the variation head model of SSAT-Adapter in the image latent space.**

*Furthest*, focuses on moving few-shot instances away from CLIP-generated anchors, increasing decision boundary separation. The second strategy, *Random*, introduces diversity by moving few-shot instances away from anchors and closer to randomly chosen instances from different classes. We extend this concept in a third strategy, *Average*, by considering all instances from different classes during adaptation. The fourth strategy, *Nearest*, focuses on adapting the few-shot instances to be closer to the nearest negative class.

As shown in Table 2, Strategy 4: *Nearest*, proved most successful. This approach ensures that adapted instances move further from anchors and closer to negative instances, strengthening decision boundary learning. By combining these elements, Strategy 4 achieves optimal expansion of the target class representation, ultimately leading to improved few-shot accuracy. The first strategy, moving few-shot instances further from CLIP-generated anchors, demonstrates accuracy approaching that of Strategy 4. While this method aims to increase decision boundary separation, it does not directly improve the model's understanding of different classes, limiting its effectiveness and demonstrating the benefit of constraining the adaptation towards the nearest negative class. In contrast, moving toward a random or average negative class results in the least accuracy gain. These two approaches inject information from contrasting categories, aiming to broaden the representation of the target class. However, the former often results in the few-shot instances moving through CLIP latent space and closer to the anchor instances, which leads to minimal information gain with high variations. The latter moves the few-shot instances to a latent space containing aggregated target information, often providing noisy information. Both ultimately hinder the model's ability to learn discriminative features, leading to lower few-shot accuracy.

**Table 3: Comparison of SSAT-Adapter with varying number of auxiliary tasks under 2-shot training setting.**

| Dataset | Number of Auxiliary Tasks | | | |
|---|---|---|---|---|
| | 2 | 4 | 10 | 20 |
| DTD | 0.513±0.002 | 0.522±0.002 | 0.524±0.002 | **0.526±0.001** |
| EuroSAT | 0.690±0.001 | 0.703±0.002 | **0.714±0.001** | **0.714±0.001** |
| ImageNet | 0.634±0.001 | 0.635±0.001 | **0.638±0.001** | 0.636±0.001 |
| Caltech101 | 0.925±0.002 | 0.933±0.001 | **0.938±0.001** | **0.938±0.001** |

*5.5.4 Number of Auxiliary Tasks.* We also investigate how the number of auxiliary tasks influences SSAT-Adapter's performance. Table 3 presents the results across various datasets under the 2-shot training setting. We observe a general trend of increasing accuracy as the number of auxiliary tasks increases, suggesting a benefit from the diverse perspectives these tasks offer during learning. However, it is important to note that the accuracy gains tend to plateau and can even slightly decrease as the number of auxiliary tasks becomes very large (e.g., 10 vs 20 tasks in our experiments). The results show a decrease in accuracy for the ImageNet dataset when the number of auxiliary tasks increased to 20, which indicates a potential risk of overfitting. When the number of auxiliary tasks increases, the generated instances become increasingly similar, making it easier for the model to memorize specific nuances of the training data rather than learning broadly generalizable representations.

## 6 LIMITATIONS & CONCLUSION

While SSAT-Adapter demonstrates significant performance improvements in few-shot image classification, there are a few drawbacks limiting its potential. The generation of anchors and auxiliary tasks introduces additional computational overhead compared to simpler fine-tuning of linear layers. Furthermore, the success of instance generation within SSAT-Adapter relies on the quality of CLIP's pre-trained representations. If CLIP's initial understanding of certain classes is limited, the potential benefits of the generated auxiliary tasks may also be reduced.

To summarize, we propose the SSAT-Adapter framework to address the challenges of few-shot image classification. Our approach leverages the pre-trained knowledge of the CLIP model to generate informative class anchors and diverse auxiliary tasks. Combined with a self-paced learning strategy, SSAT-Adapter demonstrates noticeable accuracy gains over strong baselines across a wide range of datasets. Comprehensive ablation studies highlight the importance of class anchor generation, task adaptation, and the 'easy-to-hard' self-paced auxiliary learning regime. In the future, we plan to extend SSAT-Adapter to address the challenge of handling truly unseen classes, those absent from CLIP's pre-training. Additionally, we intend to investigate the potential of SSAT-Adapter within other domains beyond image classification, including areas such as few-shot object detection or natural language generation. Furthermore, we are interested in exploring more sophisticated instance generation methods, potentially leveraging advanced generative models like diffusion or GANs.

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
