# OpenReview forum: "SSAT-Adapter: Enhancing Vision-Language Model Few-shot Learning with Auxiliary Tasks"
_acmmm.org/ACMMM/2024/Conference — MM2024 Poster_

### Official Review · Reviewer_A7nH · 2024-05-06

**Rating:** 5
**Confidence:** 3

**Summary:**

This paper proposes a new framework called SSAT-Adapter that computes image features 𝐴 and classifier weights W from the original CLIP backbone. It also generates information-rich auxiliary tasks by leveraging CLIP's language comprehension capabilities, thus enhancing CLIP's performance and adaptability for learning with fewer samples.

**Strengths:**

1. The framework utilizes auxiliary models to process image features obtained through class anchors and linear extrapolation, allowing the model to learn from multiple perspectives of the same image, thereby enhancing the overall understanding and classification accuracy of the target category.
2. SSAT-Adapter generates a series of variant image features through class anchors and linear extrapolation techniques, significantly increasing the diversity of images the model is exposed to.
3. Extensive experiments have been conducted on multiple datasets, effectively proving the efficacy of the method.

**Limitations:**

1. How do class anchors ensure they accurately represent the key features of the target category? How is the rejection sampling threshold $\delta$ determined to ensure that the generated class anchors are sufficiently distinct from other categories and representative of the target category?

2. Does SSAT-Adapter involve multiple auxiliary models and complex data processing steps which might affect memory usage and inference speed compared to CLIP-Adapter?

**Suitability:**

3

---

### Official Review · Reviewer_C3E5 · 2024-05-22

**Rating:** 5
**Confidence:** 4

**Summary:**

The paper "SSAT-Adapter: Enhancing Vision-Language Model Few-shot Learning with Auxiliary Tasks" introduces a novel framework to improve few-shot learning capabilities of the CLIP model. The SSAT-Adapter leverages CLIP's language understanding to generate auxiliary tasks that aid in learning from limited labeled data. It creates decision-boundary-focused image latents that form inter-class and itra-class instances to bridge CLIP's pre-trained knowledge with new examples and subtly expand the target classes' representation. A self-paced training regime is employed to enhance learning robustness. The proposed framework demonstrates a significant performance improvement over existing methods across eleven image classification datasets.

**Strengths:**

1. The paper presents a novel approach by leveraging CLIP's language understanding to generate auxiliary tasks, which is a unique contribution to the field of few-shot learning.
2. The use of decision-boundary-focused image latents and a self-paced training regime introduces a robust theoretical foundation for improving few-shot learning performance.
3. The paper is well-structured and clearly written, making it easy to follow the proposed methods and the results. The figures and tables effective support the text and enhance understanding.
4. The framework is thoroughly evaluated on eleven diverse image clasification datasets. The experiments show consistent performance improvements over several baseline models, including Zero-shot CLIP, Meta-Adapter, and CLIP-Adapter.

**Limitations:**

1. The generation of anchors and auxiliary tasks introduces additional computational overhead compared to simpler fine-tuning methods. This might limit the scalability and real-time applicability of the framework.
2. The performance of the SSAT-Adapter is fundamentally limited by the quality of the pre-trained CLIP model and relevance of the textual descriptions used by the CLIP model. If the pre-trained model has biases or inaccuracies in its initial representations, these issues will propagate through the SSAT-Adapter framwork, potentially limiting its effectiveness.
3. The method assumes that the data distribution in the few-shot learning scenarios is similar to what the pre-trained CLIP model has encountered. Significant deviations in data distribution can adversely affect the model's performance and are challenging to address.

**Suitability:**

3

---

### Official Review · Reviewer_3Rff · 2024-05-24

**Rating:** 2
**Confidence:** 3

**Summary:**

This paper presents SSAT-Adapter, a framework designed to enhance the few-shot learning capabilities of vision-language models, specifically the CLIP-like model. SSAT-Adapter addresses the challenge of limited labeled data in few-shot learning scenarios by leveraging CLIP's language understanding to generate informative auxiliary tasks. These tasks include inter-class and intra-class instances that bridge the gap between CLIP's pre-trained knowledge and the provided examples, expanding the representation of target classes.

**Strengths:**

The authors provide detailed descriptions of each module of their proposed method and conduct sufficient experiments.

**Limitations:**

My major concern about this paper is the method section.
1. In eq(2), if S(A_I, C) is a vector, then D(A_I) is a vector, then why "the acceptance loss" is just a number?
2. Section 4.0.1 is unclear. I can't understand why the anchor generated by rejection sampling can "represent the input’s true class", is there any former work proving this? What's the difference between this anchor and the simple class mean prototype? Why use rejection sampling instead of Gaussian Sampling?
3. In eq(1), what is the connection between the acceptance boundary width t and the anchor acceptance threshold δ? If they serve for different purposes, then why there is no description of t in the implemented details section?

Based on the current state of this paper, I can't understand why these Auxiliary Tasks can bridge CLIP's knowledge to the few-shot images. The method looks more like a data augmentation with searched hyperparameters.
The authors may consider refining the method section, maybe explaining more about the anchor generation.

**Suitability:**

2

---

### Official Review · Reviewer_fKSi · 2024-05-25

**Rating:** 3
**Confidence:** 3

**Summary:**

This paper proposes a framework, termed as SSAT-Adapter, to address the challenges of few-shot image classification. It leverages the prior knowledge of CLIP to generate informative class anchors and diverse auxiliary tasks. And this approach improves the results of few-shot image classificaiton.

**Strengths:**

1. Sufficient experiments. Enough experiments were done to verify the effectiveness of the method, and an average improvement of 2.2% was achieved on 11 datasets;
2. The writing is good, and some visualization results are drawn to analyze the method.

**Limitations:**

I do not perceive the methodology employed here as groundbreaking; rather, it resembles a combination of elements A plus B with an addition of C, albeit yielding favorable outcomes. Moreover, the paper's introduction feels insufficiently detailed, lacking depth in explaining the motivation behind the research. I hope you could succinctly articulate your motivation and highlight the novelty of your genuinely innovative approach. Should this be satisfactorily demonstrated, I might reconsider my evaluation.

**Suitability:**

3

---

### Meta-Review · Area_Chair_pkoS · 2024-07-02

**Recommendation:** Accept (Poster)
**Confidence:** 3

**Metareview:**

This paper received mixed final ratings after the rebuttal period. After checking the paper, reviews, and rebuttal document, the AC tends to support  Reviewers C3E5 and A7nH. The overall quality of this paper is satisfactory and the rebuttal document addressed most of the concerns. Therefore, I recommend accepting this paper.